# Response Length Perception and Sequence Scheduling: An LLM-Empowered LLM Inference Pipeline

**Zangwei Zheng**[1], **Xiaozhe Ren**[2], **Fuzhao Xue**[1], **Yang Luo**[1], **Xin Jiang**[2], **Yang You**[1]

[1]Department of Computer Science, National University of Singapore
[2]Noah's Ark Lab, Huawei.

{zangwei, f-xue, yangluo, youy}@comp.nus.edu.sg; {renxiaozhe, jiang.xin}@huawei.com
https://github.com/zhengzangw/Sequence-Scheduling

## Abstract

Large language models (LLMs) have revolutionized the field of AI, demonstrating unprecedented capacity across various tasks. However, the inference process for LLMs comes with significant computational costs. In this paper, we propose an efficient LLM inference pipeline that harnesses the power of LLMs. Our approach begins by tapping into the potential of LLMs to accurately perceive and predict the response length with minimal overhead. By leveraging this information, we introduce an efficient sequence scheduling technique that groups queries with similar response lengths into micro-batches. We evaluate our approach on real-world instruction datasets using the LLaMA-based model, and our results demonstrate an impressive 86% improvement in inference throughput without compromising effectiveness. Notably, our method is orthogonal to other inference acceleration techniques, making it a valuable addition to many existing toolkits (*e.g.* FlashAttention, Quantization) for LLM inference.

## 1 Introduction

Large language models (LLMs) [2, 6, 15, 19] have transformed the field of natural language processing (NLP) and have demonstrated remarkable success in various NLP tasks such as language translation [23], question-answering [28], and text summarization [38]. However, the deployment of LLMs at scale poses challenges due to their prohibitively expensive inference cost [1, 27]. The computational resources required to process millions of queries, as is the case with currently deployed LLMs like ChatGPT [19], are substantial. As a result, reducing the inference cost of LLMs has become a crucial research direction in recent years.

In real-world scenarios, the lengths of responses to various queries exhibit significant variability. As depicted in Figure Figure 2a, although different models display slightly diverse response length distributions, a common pattern emerges with the presence of response lengths across a wide range. Consequently, when performing large language model (LLM) inference in batches, the inclusion of sequences with differing response lengths leads to inefficiencies. Shorter sequences are forced to wait for longer ones to complete, resulting in computational waste. This issue is depicted on the left side of Figure Figure 1, where redundant tokens account for a substantial portion (66%) of the overall tokens generated. Given the quadratic time complexity of inference, such inefficiencies impose a significant burden on the inference process.

Humans possess the ability to estimate the length of an answer to a question based on their understanding of the query. For instance, questions like "What is the capital of France?" typically elicit shorter responses compared to inquiries such as "Can you explain the history of the French Revolution?" Intriguingly, we observe that LLMs fine-tuned for instruction comprehension, such as ChatGPT and Claude, also exhibit a certain degree of response length perception. Moreover, even smaller models

37th Conference on Neural Information Processing Systems (NeurIPS 2023).

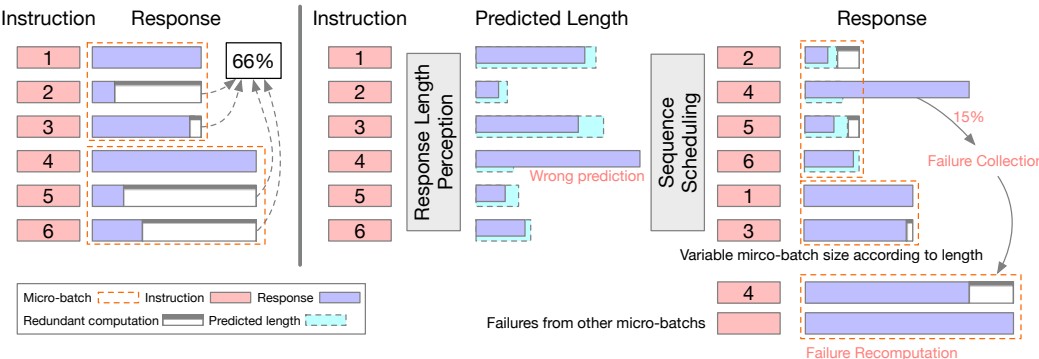

Figure 1: **Left:** Vanilla batch inference leads to underperformance with 66% redundant tokens when short and long responses are in the same batch. **Right:** The pipeline of our sequence scheduling. First, the response length perception module estimates the response length for each instruction. Sequence scheduling groups instructions with similar predicted lengths together and larger batch sizes for shorter responses. Failure collection and recomputation strategy is adopted to avoid wrong predictions degenerating the performance.

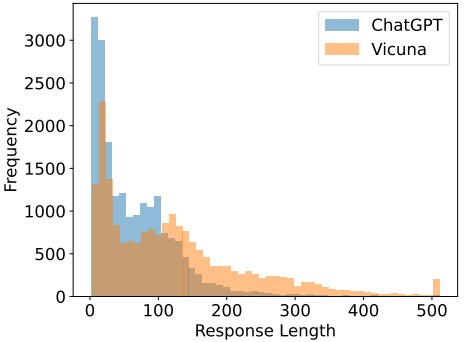

(a) Response length distribution of 10k instructions from ChatGPT and Vicuna. Response lengths larger than 512 are truncated.

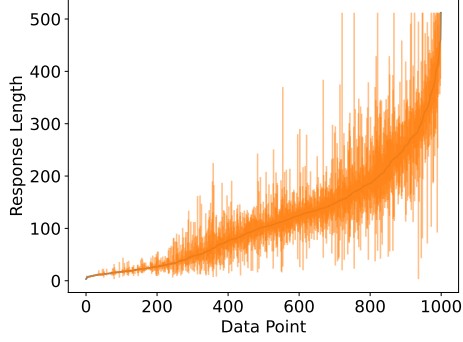

(b) Distribution of mean length among 3 times generations on 1k instructions. Error bar denotes maximum and minimum length in generations.

Figure 2: Distribution of response length and variance. The response length is measured in tokens.

like LLaMA-7B, when instruction-tuned on a length prediction dataset, can acquire this capability. Despite the variability in output length under multiple sampling, as demonstrated in Figure Figure 2b, these models achieve impressive performance in perceiving the length of their responses.

Leveraging the response length perception ability of LLMs, we can employ it to enhance the scheduling of instructions within micro-batches. This represents an example of software-hardware co-design, where the inherent capabilities of LLMs contribute to the acceleration of the LLM inference process. Our proposed sequence scheduling system intelligently groups queries based on their perceived response lengths, effectively minimizing computational waste. To further improve efficiency, we introduce a failure collection and recomputation strategy, as well as a variable batch size approach. These additional techniques complement the sequence scheduling system and contribute to further improvements in inference throughput.

In order to evaluate the effectiveness of our proposed approach, we conduct experiments on real-world instruction datasets using Vicuna [4], an instruction-tuned LLaMA model [34]. Our length predictor module surpasses the performance of previous methods in accurately estimating response lengths. Our sequence scheduling system demonstrates a remarkable improvement in inference throughput, achieving an 86% increase compared to the original inference process, all while maintaining performance quality. These results highlight the potential of sequence scheduling with response length

perception as a valuable addition to the existing toolkit (*e.g.* Flash Attention [7], Quantization [8, 11, 35]) for large language model inference.

To summarize, our contributions are as follows:

- We investigate the response length perception ability of LLMs and demonstrate that instruction tuning can enhance this capability.

- We introduce a novel LLM inference pipeline called sequence scheduling that leverages LLMs' response length perception. This approach intelligently groups queries with similar response lengths, reducing computational waste and improving inference throughput without compromising performance.

- We present comprehensive experimental results on real-world instruction datasets using the Vicuna-7B model. Our proposed method achieves an impressive 86% improvement in inference throughput compared to the original inference process.

## 2  Related Work

**Large Language Model As-a-service.**  Large Language Models (LLMs) [2, 6, 15, 19] have been successful in building strong foundation models by scaling language models to a large scale. With instruction tuning [25], LLMs can align with human requirements and provide them as a service for practical usage. Currently, LLMs such as ChatGPT [19] and PaLM [6] have been deployed in Bing and Bard as a service and perform a significant amount of inference every day. Therefore, reducing the inference cost of LLMs is a crucial research direction.

**Efficient LLM Inference.**  In recent years, there has been increasing interest in developing efficient inference techniques for large language models (LLMs) [18]. Kernel fusion [5, 7] involves the use of highly optimized kernels to reduce memory access and improve computation speed. Parallelism methods, such as pipeline parallelism [17, 30] and tensor parallelism [22, 30], have been used to distribute the workload across multiple GPUs, enabling efficient scaling of LLM inference. Quantization [8, 11, 35] has also been explored as a means of compressing the parameters of LLMs for efficient inference. In addition to these methods, there has been some work on optimizing batch processing for LLMs [3, 10, 29]. For example, [3] focused on batchifying queries in few-shot settings, while [10] proposed grouping sentences into batches based on input length. For LLM, the cost of generation exceeds the forward of prompts. Our method focus on the generation process and group sentences according to the predicted output length.

**Response Length Prediction.**  The previous work on response length prediction has primarily focused on non-auto-regressive generation (NAR) translation tasks [14]. In these tasks, the entire sentence is generated at once, so predicting the length of the response is crucial. Various techniques have been proposed to address this problem. For instance, [32] proposed a simple approach based on the statistics of the dataset and a bias term, while [13] predicted the number of tokens each input token would be translated into. Some methods, such as [9, 12], added a special [LENGTH] token to the encoder, while others, such as [20, 21, 24, 31], used a pooling layer and MLP classifier to predict the response length based on the encoder's outputs. However, these methods are primarily applicable to machine translation tasks, where the target sequence length is similar to the source length and thus easier to predict. In contrast, our proposed approach is specifically designed for large language model inference tasks, where the types of queries and their corresponding response lengths vary widely.

## 3  Response Length Perception

### 3.1  Perception in Advance (PiA)

Instruction-tuned LLMs have shown the ability to align with human understanding and provide helpful and safe responses. Interestingly, we have found that these models possess an overall understanding of the entire response they are going to generate, similar to how humans formulate their responses. In our experiments, we asked these models to predict the length of the responses they were about to generate, even though no explicit task for response length perception was included during pretraining.

Table 1: Examples of generation with Perception in Advance (PiA). Instruction-learned models are able to follow the PiA instruction. Small models (Vicuna) may fail to generate the response.

| |
|---|
| **Prompt:** |
| Create a fun math question for children. |
| Before responding to the above instruction, you have to predict the length of your response. Print the estimated number of words in your response in the first line. Then change to a new line to respond to the instruction. |
| **GPT-4:** |
| Estimated response length: 60 words. |
| Sure, here's a fun math problem: There are 7 apples in a basket. A friendly squirrel comes and $\cdots$ |
| **ChatGPT:** |
| 4 words |
| If you have 10 apples and give 3 to your friend, how many do you have left? <EOS> |
| **Claude:** |
| 225 |
| Here is a fun math question for children: Jenny the Juggler was practicing her act. In one hand she held $\cdots$ |
| **Vicuna:** |
| 101 <EOS> |

Table 2: Performance of response length perception via Perception in Advance across different LLMs.

| | Perception in Advance (PiA) | | | Perception Only (PO) | | | |
|---|---|---|---|---|---|---|---|
| | Error(w) ↓ | Acc-50 ↑ | Acc-100 ↑ | Error(w) ↓ | Acc-50 ↑ | Acc-100 ↑ | Failure ↓ |
| GPT-4 | 22 | 80% | 100% | 100 | 28% | 55% | 0% |
| ChatGPT | 51 | 77% | 90% | 89 | 55% | 68% | 2% |
| Claude | 37 | 64% | 96% | 63 | 52% | 84% | 0% |
| Bard | 70 | 44% | 72% | 130 | 28% | 50% | 28% |
| HugginChat-30B | 77 | 52% | 72% | 113 | 56% | 72% | 12% |
| Vicuna-13B | 94 | 49% | 73% | 92 | 55% | 75% | 0% |
| Vicuna-7B | 123 | 40% | 65% | 122 | 40% | 65% | 0% |

As shown in Table 1, we introduce a modification to the original prompt by including an instruction to estimate the response length in advance. We refer to this method as Perception in Advance (PiA). We applied this modified prompt to various LLMs and observed their responses to the instruction.

**Comprehension of Response Length.** Our experiments demonstrate that instruction-tuned LLMs possess a strong comprehension of response length estimation when provided with the PiA instructions. To evaluate the effectiveness of the PiA method, we conduct tests using 175 alpaca seed instructions [33]. The results, shown in Table 2, indicate that GPT-4, ChatGPT, and Vicuna successfully followed the instructions for all queries.

One interesting aspect we discovered during our analysis is that LLMs exhibit a better understanding of words compared to tokens. This observation aligns with how humans comprehend and generate responses. By considering words as the unit of measurement, we can obtain more reliable estimates of response length.

To quantitatively evaluate the accuracy of response length perception, we employed the metric Error(w), which measures the difference between the estimated number of words and the actual word number. We also consider two thresholds for accuracy: Acc-50 and Acc-100, where a prediction is considered correct if the difference falls below 50 and 100 words, respectively. Our analysis reveals that both GPT-4 and Claude exhibited exceptional performance in response length estimation. They achieve an error of fewer than 50 words and demonstrate an Acc-100 score exceeding 90%. These results demonstrate the high accuracy and reliability of the response length predictions generated by these models. Furthermore, it is worth noting that models with a higher number of parameters exhibit superior performance in response length prediction.

**Side Effect of PiA on Response Generation.** Introducing EiA has a side effect on the response generation process. First, since the estimated length is visible to the model during the generation phase, it can influence how the model generates responses. The model can perceive the estimated

Table 3: Response length perception performance comparison: we evaluate different prediction methods for vicuna model inference length on 10k instructions.

| | Error ↓ | Acc-50 ↑ | Acc-100 ↑ |
|---|---|---|---|
| **GPT-2 (1.5B)** | | | |
| Pooling + MLP | 127 | 35% | 53% |
| [LEN]-token Fine-tune | 92 | 43% | 64% |
| **LLaMA-7B** | | | |
| Pooling + MLP | 127 | 35% | 53% |
| [LEN]-token Fine-tune | 81 | 46% | 70% |
| **Vicuna-7B** | | | |
| Pooling + MLP | 73 | 55% | 75% |
| [LEN]-token Fine-tune | 84 | 47% | 72% |
| Perception Only | 193 | 38% | 59% |
| Instruction Tuning | **63** | **56%** | **81%** |

length as a constraint and attempt to tailor its response to fit the predicted length. This behavior can be seen as a length-limited generation.

To investigate the impact of PiA on response generation, we compared the error in response length perception between the PiA method and the Perception Only (PO) method. In this case, we compare the response length of unmodified instructions with the perceived length values. Note that the response length can vary across different sampling generated from the same instruction. Figure 2b illustrates the variability in response length for 1,000 data points, highlighting the wide range of possible lengths. As a result, there is no definitive "ground truth" for response length but rather a range of possible lengths. To simplify the estimation task, we aim for the model to predict the maximum potential length, as using the mean length has limitations, as discussed in the subsequent section.

For smaller LLMs, such as Vicuna 7B and 13B, we observed that they almost ignore the estimated length. On the other hand, GPT-4 and Claude demonstrate a stronger tendency to tailor their answers to fit the estimated length, resulting in significantly smaller error numbers.

In addition, we observed that introducing the PiA method might negatively impact the response quality for smaller LLMs. For instance, in the case of Vicuna-7B, we notice instances where the model failed to generate a response after predicting the length. This behavior can be attributed to the limited capacity of smaller LLMs to handle multiple tasks simultaneously.

**Perception Only is Harder.** In order to address the side effects associated with response generation influenced by estimated length, we adopt Perception Only (PO) style for sequence scheduling since it decouples the prediction and generation processes.

One straightforward approach to perform PO is to employ the same prompt but solely retrieve the length prediction. We then compare this estimated length with the actual response generated using the original prompts. As presented in Table 2, it is evident that although LLMs are still capable of estimating the length, their error rates are significantly higher, and accuracy scores are lower compared to the Perception in Advance (PiA) approach.

## 3.2 Instruction Tuning

While the Perception in Advance approach may be sufficient for GPT-4 in terms of small side effects on the generation process and enabling sequence scheduling, we aim to completely decouple the prediction and generation stages to avoid any potential influence of the estimated length. Additionally, we want to empower smaller models with the ability to accurately perceived response lengths. To achieve these objectives, we employ instruction tuning [25].

During the instruction tuning phase, we utilize a modified prompt format that prompts the model to predict the length of the response instead of generating the entire response. We select a subset of 10,000 prompts from the alpaca dataset [33]. We sample four generations for each prompt and set the target length as the maximum one. The target text is a number only, so the generation cost

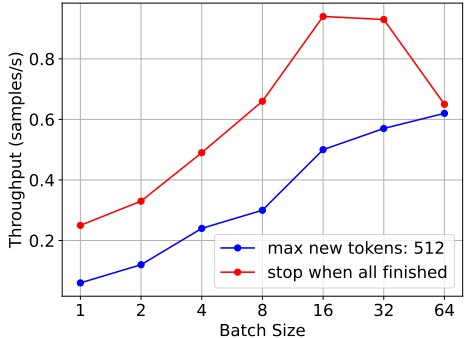
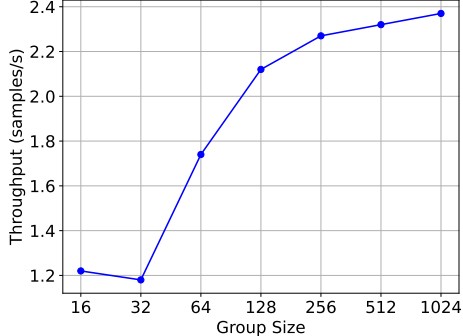

(a) When generating a fixed length (blue), throughput grows almost linearly. When generating until finishing in a batch, long response in large batch size degrades performance.

(b) Larger group size makes scheduling more effective. Scheduling fails when the group size is too small (32). The improvement of more instructions in a group decreases as group size grows.

Figure 3: Throughput vs. batch size and group size.

in the prediction process is minimized. To optimize the training process and reduce computational resources, we employ the efficient training method LoRA [16], which requires negligible memory compared with the LLM, and train the model for three epochs. For further details on the experimental setup, we provide comprehensive information in the appendix section of this paper.

Table 3 presents the experimental results, showcasing the improvement achieved through instruction tuning. The prediction error is greatly reduced from 193 to 63, and Acc-50 shows an improvement of 18%. We also compare our instruction tuning approach with previous length prediction methods utilized in NAR generation, such as fine-tuning with a length token and employing an MLP to classify pooled hidden states. Although these methods also exhibit performance improvements, they fall short compared to the effectiveness of instruction tuning. These results highlight the model's ability to comprehend and effectively utilize the provided instruction. Furthermore, when using alternative models such as LLaMA-7B or smaller models like GPT-2 to generate length predictions for Vicuna, the pooling + MLP approach fails completely, and fine-tuning with a length token falls short when compared to Vicuna's self-prediction capabilities.

## 4 Sequence Scheduling

### 4.1 Method

Having established an accurate response length perception module, we can now leverage it to enable sequence scheduling for efficient inference. As illustrated in the left side of Figure 1, when instructions with highly disparate response lengths are batched together, significant, redundant computations occur, resulting in reduced inference throughput. Therefore, by grouping instructions with similar response lengths together, we can accelerate the inference process.

Before delving into the specifics of sequence scheduling, it is important to understand the significance of inference with large micro-batch sizes (mbs). As depicted in the Figure 3a, deploying Vicuna on an 80GB A100 GPU highlights the benefits of larger batch sizes in leveraging the parallel computing power of the GPU. When generating a fixed number of tokens for each sample, the throughput exhibits almost linear improvement up to a batch size of 16, after which the rate of improvement slows down. On the other hand, if the generation of each batch halts when all samples have finished generating their responses, the throughput also increases linearly for batch sizes smaller than 16, with a higher ratio than the fixed-token approach. However, as the batch size continues to increase, performance begins to decline. This is due to the fact that larger batch sizes have a high probability of entailing a longer response length, resulting in significant redundant computations.

To enable efficient sequence scheduling, we make the assumption that the number of instructions to process at a time (group size) is larger than the micro-batch size for a single GPU, which holds true given the widespread usage of LLMs. While a straightforward approach for sequence scheduling is to sort the instructions by their predicted length and split them into batches for processing, we

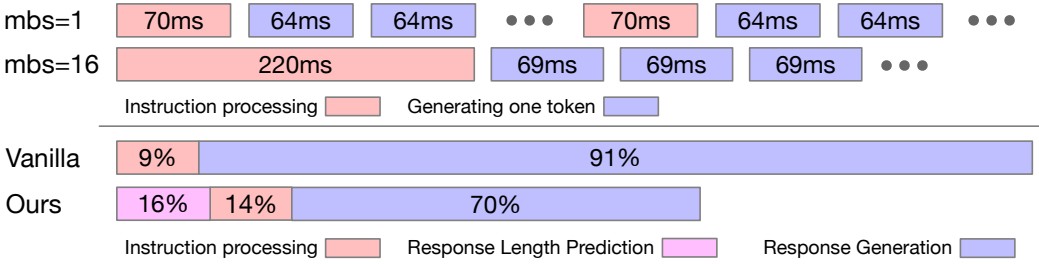

Figure 4: **Top:** Time comparison between instruction processing and token generation. Instruction processing only takes time to generate a few tokens. **Bottom:** Percentage distribution bar for different components of vanilla and our methods.

explore additional designs to further accelerate throughput. The pipeline of our method is depicted on the right side of Figure 1.

**Failure Collection and Recomputation (FCR)**  Although the length predictor achieves a reasonably high accuracy of 81% (acc-100), there is still a chance that some samples have predictions that deviate significantly from the true response length. These incorrect predictions can greatly disrupt the efficiency of batch processing. For example, if a long response is mistakenly predicted as a short one and included in a batch with predominantly short responses, the overall processing time is affected as the short queries are forced to wait for the completion of the long one. To mitigate this issue, we implement a mechanism called Failure Collection and Recomputation (FCR). We restrict the number of newly generated tokens to be at most the maximum predicted length within a batch. Any instruction that exceeds this predicted length is considered a failure and is collected separately for further recomputation at the end of a group size inference process. Given the relatively low failure ratio, this approach enables faster generation of shorter responses with limited time spent on regenerating failed instructions.

**Variable Batch Size (VBS)**  One important consideration in sequence scheduling is memory usage during response generation. Shorter responses require less memory compared to longer ones. However, if we allocate a larger batch size without considering the possibility of misclassified long responses as short ones, we might encounter out-of-memory errors. With the help of Failure Re-collection, we introduce the Variable Batch Size (VBS) technique. We allocate a larger batch size for shorter responses. A simple rule is to maintain the same number of tokens in each batch. Given the baseline batch size $B_0$ corresponding to a specific length $L_0$, we adjust the batch size based on the desired length $L$ using the formula $B_0 \cdot L / L_0$. This approach optimizes memory usage by allowing larger batch sizes for shorter responses while preventing memory overflow caused by RC.

**Binning and Predicting the Max Length**  In training the length predictor, we employ a binning strategy that categorizes the target length into bins. In our approach, we use bins with a cell size of 50 and round the numbers to the nearest bin that is greater than the actual length. First, the objective of predicting the bin number is simpler and easier as it requires only an approximate estimation. Second, binning provides a buffer for potential errors in length prediction. Even if the actual length deviates within the same bin, it does not impact the effectiveness of our methods.

In addition, we choose to predict the maximum length of four times the generation process because the consequences of underestimation are more severe compared to overestimation. By predicting the maximum length, we ensure that the generated responses have enough capacity to accommodate even the longest possible output. Predicting the mean length, on the other hand, would result in a higher failure re-collection ratio, as it may underestimate the length for some queries, leading to potential disruptions in the batching process.

**Overhead of Length Prediction**  Given that we must predict the lengths of all instructions within a group before generating their responses, there is an inherent overhead associated with response length prediction. This overhead entails calculating the keys and values for the instruction tokens and generating a small number of tokens (typically 2 to 3 tokens). However, as depicted in Figure 4, the processing time for instructions is extremely fast, requiring only the time to generate a few tokens

Table 4: Performance of sequence scheduling with different response length perception module.

| | Throughput (samples/s) ↑ | Improvement ↑ | Tokens/batch ↓ |
|---|---|---|---|
| Vanilla | 1.22 | | 377 |
| Ground Truth Preditor | 2.52 | +107% | 201 |
| Pooling + MLP | 1.96 | +61% | 216 |
| [LEN]-token Fine-tune | 2.10 | +72% | 210 |
| Perception Only* | 1.40 | +15% | 328 |
| Instruction Tunning (mean) | 1.77 | +45% | 211 |
| Instruction Tunning (max) | **2.27** | **+86%** | **208** |

Table 5: Ablation study on three components of sequence scheduling method.

| Binning | FCR | VBS | Throughput (samples/s) ↑ | Improvement ↑ | Tokens/batch ↓ |
|---|---|---|---|---|---|
| × | × | × | 1.73 | +42% | 271 |
| × | ✓ | × | 1.76 | +44% | 222 |
| × | ✓ | ✓ | 2.22 | +82% | 209 |
| ✓ | × | × | 1.58 | +30% | 300 |
| ✓ | ✓ | × | 1.86 | +52% | 209 |
| ✓ | ✓ | ✓ | **2.27** | **+86%** | **203** |

(ranging from 1 to 4 tokens). Consequently, this overhead can be effectively offset by the overall acceleration achieved through sequence scheduling.

## 4.2 Experiments

**Experimental Setting.** Our experiments are conducted on two datasets: a set of 10,000 prompts from a subset of the alpaca dataset [33] (which is different from the one used to train the length predictor) and a set of 429 prompts from the Instruction-in-Wild datasets [36]. The former consists of self-instructed prompts, while the latter contains real-world collected prompts.

For our baseline experiments, we set the batch size to 16. Regarding the variable batch size strategy, we use a batch size of 16 for instructions with a length ($L$) greater than or equal to 300. For instructions with a length below 300, we calculate the batch size using the formula $256 \times 50 / L$ and then round it to the nearest power of 2 for better GPU utilization. We maintain a fixed group size of 256. The inference is performed on the Vicuna-7B [4] model using an 80GB A100 GPU. We sample generations with a temperature of 0.5 for diversity in responses.

We evaluate the throughput at the sample level and report the number of samples processed per second and the corresponding improvement compared to the vanilla baseline, where inference is conducted with batch size 16. The average length refers to the mean length of each batch, where a smaller average length indicates reduced redundant computation.

**Results.** Table 4 presents the performance of sequence scheduling with different response length perception modules. Among the length predictors considered, the one with instruction tuning demonstrates the best performance, achieving an impressive 85% improvement in throughput. The acceleration is due to a smaller number of tokens per batch, which includes wasted tokens due to short sentences waiting for the long ones to be completed. It is important to note that the estimation-only method exhibits a significantly higher failure re-collection ratio, resulting in reduced performance. Hence, we report the results obtained using the vanilla sequence scheduling approach.

When predicting the mean length, the failure re-collection ratio increases to 29.8%, which is considerably higher compared to the 15.3% achieved when predicting the maximum length. Consequently, the performance improvement drops to 45%. Alternatively, if we utilize the ground truth (i.e., the maximum length observed during multiple inferences) as the length predictor, it serves as an upper bound for this method. Our approach performs only 0.25 samples/s slower than the upper bound, showcasing that an effective length predictor can yield substantial improvements.

Table 6: Performance of sequence scheduling on Instruction-in-Wild dataset.

| | Throughput (samples/s) ↑ | Avg. length ↓ | Error ↓ | Acc-50 ↑ | Acc-100 ↑ |
|---|---|---|---|---|---|
| Vanilla | 0.78 | 475 | – | – | – |
| Estimation Only | 1.07 | 422 | 358 | 20% | 38% |
| Instruction Tunning | 1.24 | 299 | 139 | 43% | 68% |

Table 7: Study on the Failure Collection and Recomputation (FCR) ratio.

| k | Avg. Wait | Max. Wait | FCR ratio | FCR time | Throughput (samples/s) |
|---|---|---|---|---|---|
| -50 | 42% | 63% | 34% | 21% | 1.80 |
| -10 | **37%** | 53% | 20% | 20% | 2.10 |
| 0 | **37%** | **52%** | 15% | 20% | **2.27** |
| +10 | 45% | 52% | 14% | 14% | 2.20 |
| +50 | 38% | 55% | 7% | 7% | 2.10 |

Furthermore, our method exhibits a low variance in throughput, with a value of 0.05 over three times experiments. This indicates the stability and consistency of our approach in achieving improved inference speed.

Table 5 presents an ablation study on the three components of our approach, demonstrating their individual contributions to the overall improvement. We observe that Binning enhances the effectiveness of Variable Batch Size (VBS), leading to a more powerful combination. Each component plays a significant role in achieving the final improvement in throughput. Furthermore, Table 6 showcases the performance of our methods on the Instruction-in-Wild dataset, confirming the effectiveness of our approach across different datasets.

In addition, on the right side of Figure 3b, we analyze the relationship between throughput and group size. We observe that a larger group size provides more flexibility for sequence scheduling, resulting in improved throughput. However, the rate of improvement gradually slows down as the group size increases. Thus, it becomes crucial to strike a balance between the overhead involved in gathering the group size information in real-world scenarios and the achieved improvement.

**Waiting time.** Apart from the throughput, another important metric is the waiting time for each user. We introduce three metrics for comparison: maximum waiting time, average waiting time, and their ratio (maximum waiting time multiplier). The maximum timing time is a fairness metric for measuring individual request delay. We define the waiting time for a user to be the delay from receiving the request (start of processing a group) to generate the corresponding response. With group size 256, our method saves 63% average waiting time compared to the vanilla. Due to the FCR mechanism, our maximum waiting time multiplier is 2.7, which is 1.4 times the vanilla's one. However, with inference speed acceleration, the maximum wait time is also reduced by 48%.

**FCR ratio.** One factor influencing inference latency is the Failure Collection and Recomputation (FCR) ratio. This ratio represents the proportion of FCR samples recalculated at the end of a batch, which causes delays in processing. To assess the effect of different FCR ratios, we modify the predicted length by a constant value $k$. A larger predicted length results in a lower FCR ratio (more tolerable). However, a long response with a short predicted length may introduce more waste generation in batch (generating $k$ more times). The experimental results are as shown in Table 7. The Avg. Wait and Max. Wait columns denote the the average and maximum wait times compared to the vanilla method. FCR time indicates the proportion of time utilized for FCR processing.

When using an accurate length predictor, it is advisable to directly utilize the predicted length with a 15% FCR ratio. This approach strikes a balance between the time spent on FCR recomputation and the time wasted during batch computation. In some cases, the generation process might require just a few more tokens to complete the task, but it is difficult to determine which response will finish or if it will finish even after generating more tokens. This concern is also the reason why the maximum length among four times generation is used, as it aims to reduce the potential FCR ratio.

# 5 Limitation and Discussion

One limitation is that accurate response length prediction is not always guaranteed, even with the instruction-tuned length predictor module we developed. While our experiments demonstrate promising results, there is still room for improvement in the accuracy of response length estimation.

Besides, although sequence scheduling significantly reduces redundant computations, it cannot completely eliminate the issue. Even with a ground truth response length predictor, the ratio of redundancy decreases from 66% to 33%, leaving ample room for further improvement. Recent works such as ORCA [37] have proposed novel inference systems that offer alternative solutions to mitigate the redundant computation problem.

Another limitation of our work is that we focus less on the process of input instructions. As the maximum token length supported by LLMs increases, users may input very long instructions, leading to redundant computations and inefficiencies. Future work could explore a combination of our proposed sequence scheduling with input batch scheduling techniques, such as those discussed in Fang et al. (2021) [10], to further optimize the inference process.

Our approach assumes that the group size exceeds the capacity of a single GPU. As large language models may become a ubiquitous infrastructure, similar to search engines, the number of queries will increase significantly. Furthermore, the emergence of models like GPT-4 with 32k sequence length support and Claude with 100K sequence length support amplifies the challenge of handling varying response lengths, highlighting the relevance and effectiveness of our method.

Our approach is easily extensible to multi-GPU settings, where multiple GPUs are used for faster processing and handling larger model sizes. By reallocating batches of sequences with different perceived response lengths to different GPU nodes, our method remains effective in high-performance computing environments. This extension ensures scalability and maintains the efficiency gains achieved in our approach.

Some argue our method needs to reorder inputs. However, the order of input sequences is not critical; rather, the focus is on batch assembly. If requests in a group arrive together, the overhead from reordering indices is negligible. The main latency arises from the response length perception, not index reordering. After forming batches, their processing order is flexible. Although this might affect individual user latency, it's bound by a group's processing time. Batch shuffling can also offset order impacts. In multi-GPU setups, batches with varied predicted response lengths are sent to different GPUs for simultaneous processing, emphasizing batch assembly over sequence order.

For more powerful LLMs (such as GPT-4, Claude, etc.), our proposed PiA method introduces no need for instruction-tunning and introduces no overhead during sequence-scheduling. This can better handle long inputs and multi-turn conversation. However, as these models are not open-sourced, it is unable to perform experiments on them. Detailed dicussion on PiA method can be found in appendix C.

Our findings indicate that large language models (LLMs) possess a comprehensive comprehension of their generated responses. This understanding opens up possibilities for the creation of faster inference techniques, including non-autoregressive methods, that can overcome the performance constraints associated with sequential token generation.

# 6 Conclusion

In this paper, we proposed a novel technique called sequence scheduling, which optimizes large language model (LLM) inference by leveraging response length prediction. Our approach groups queries with similar response lengths in batches, reducing computational waste and improving inference throughput. We introduced the concepts of failure re-collection and variable batch size to further enhance efficiency. Experimental results on real-world instruction datasets using the Vicuna-7B model demonstrated an 86% improvement in throughput without sacrificing performance quality. Our method offers a valuable addition to the existing toolkit for LLM inference, addressing the challenge of efficient deployment of LLMs at scale.

## Acknowledgements

Yang You's research group is being sponsored by NUS startup grant (Presidential Young Professorship), Singapore MOE Tier-1 grant, ByteDance grant, ARCTIC grant, SMI grant and Alibaba grant. This work is sponsored by Huawei Noah's Ark Lab. We would like to thank **Shenggan Cheng** and **Ziming Liu** for their valuable discussion, including advice on the implementation.

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

# Appendix

## A  Additional Instruction Tuning Implementation Details

To fine-tune the instruction-tuned length predictor module, we followed a specific procedure. First, we prepared the dataset by sampling each instruction four times. One sample was obtained using greedy decoding with a temperature of 0, while the other three samples were generated using a temperature of 1.0 with different random seeds. The maximum length among the four samples was used as the target length for each instruction.

For the training dataset, we constructed new instructions by appending the requirement of perceiving the response length only. The prompt we used for this purpose was: "Don't output the response for the above instruction. Instead, you need to predict the number of tokens in your response. Output only one number."

During the training process, we employed the same hyperparameters as the Vicuna [4] instruction tuning process on LLaMA [34]. Specifically, we set the learning rate to 0.00005 and trained the model for three epochs. We applied the LoRA [16] method solely to the query and key linear layer. The training was conducted on a single 80GB A100 GPU. All codes are implemented in PyTorch [26].

## B  Distribution of Instruction-in-Wild Dataset

The histogram of response length on Instruction-in-Wild [36] dataset is shown in Figure 5. The response length is more diverse and contains more responses with a long length.

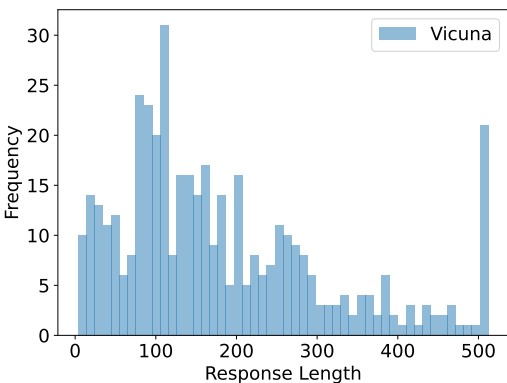

Figure 5: Response length distribution of Instruction-in-Wild dataset.

## C  Discussion on Sequence Scheduling with PiA

In the main text, we presented the sequence scheduling technique using instruction-tuned models, where the LoRA weight was utilized for response length perception. However, recent LLMs such as GPT-4 and Claude have shown promising performance in Perception in Advance (PiA), which allows them to leverage PiA for sequence scheduling without the need for additional weights.

To understand the PiA process, consider the following format:

[Input] [PiA Prompt] [PiA Response] [Response]

Here, the PiA prompt might be something like "Please estimate the length of your prediction," and the subsequent PiA Response could be "The estimated response length is [X]". The primary overhead of this method involves the generation of the aforementioned tokens, typically no more than 30 in number.

As stated section 3.1, PiA may have different behaviour than Perception Only (PO) because the response is now depend on PiA prompt and response. Howver, we can eliminate this effect due to the fact that the input and the response can be detached from the PiA Prompt and its Response.

To ensure the quality of the response remains unchanged, we can use the following two methods:

- After the generation of the PiA response, we can exclude both the PiA Prompt and PiA Response embeddings. The generation of the primary Response would then only use the kv-cache of the input.
- Alternatively, the Response can be generated with both the PiA Prompt and PiA Response masked.

Both these strategies ensure the preservation of generation quality. In our experiments with models like ChatGPT and other leading-edge Large Language Models (LLMs), merely invoking the API didn't allow for the bypassing of the PiA prompt's influence. However, this implementation is quite simple for the companies. This observation also led us to introduce the Perception Only (PO) approach, wherein we compared the mean predicted length against the original prediction length as an estimation of performance of PiA method in practice.

To further improve the speed of inference in this setting, one potential approach is to reuse the key-value (kv) cache of input queries from the response length perception stage during the inference stage. This eliminates the need for recomputing the kv-cache on input instructions, thereby saving valuable processing time.

One strategy we explored is offloading the kv-cache to the main memory and then loading it back into the GPU memory. However, the transfer time between the CPU and GPU can be significant and greatly impact overall performance, often surpassing the time saved by recomputation. To address this, one possible improvement is to offload and load only a portion of the kv-cache asynchronously, reducing the transfer time overhead. This is an area that we leave for future work and exploration.

Another approach we investigated involved compressing the kv-cache and storing it directly in the GPU memory. We applied FlexGen's quantization method [29] for compression and found that it had minimal impact on performance. However, this approach does consume additional memory and can lead to a smaller batch size, potentially degrading overall performance. A potential avenue for further exploration is to combine compression and offloading techniques to strike a balance between memory usage and performance.

Considering these factors, we have chosen to continue using the recomputation strategy for the PiA response length perception module in our proposed pipeline. While there is potential for optimizations through offloading and compression, further investigation and refinement are required to achieve substantial performance gains in practice.

