# OpenReview forum: "Response Length Perception and  Sequence Scheduling: An LLM-Empowered LLM Inference Pipeline"
_NeurIPS.cc/2023/Conference — NeurIPS 2023 poster_

### Official Review · Reviewer_Nixu · 2023-06-27

**Soundness:** 3 good
**Presentation:** 3 good
**Contribution:** 3 good
**Rating:** 6
**Confidence:** 4

**Summary:**

The authors propose a simple method to perceive the length of an LLM response by asking the LLM. Then, the authors propose to groups queries with similar response lengths into micro-batches, which are then allocated to different GPU nodes and processed in parallel . The authors show empirical gain in terms of throughput. This approach is also orthogonal to other inference acceleration approaches.

**Strengths:**

1. The proposed method is simple and exact and does not sacrifice response quality while achieving speedup.

2. The authors show impressive speedup in terms of throughput.

3. The speed up the author achieve can be applied on top of other inference acceleration techniques.

**Weaknesses:**

1. One fundamental weakness of the paper lies in the assumption that requests can be reordered, which may not hold in production. The author did not show any fairness metric which may help readers understand how their method affect each individual request in practice.

2. The author did not show any fine-grained ablation studies examining how often and how often the requests have been reordered and how it affects inference latency.

I would happily raise my rating if the authors can present more thorough ablation studies.



**Questions:**

I have listed my concerns above.

**Limitations:**

The authors adequately addressed the limitations and potential negative societal impact of their work.

---

> ### Author Rebuttal · Authors · 2023-08-05
>
> We sincerely thank the reviewer for the insightful comments. We are pleased to see that the reviewer acknowledges our contribution. The questions are answered below.
>
> **Weakness-1**: The foundation of our method is not based on the assumption that individual requests can be "reordered," but rather on the expectation that a "group" of requests is received simultaneously (as mentioned on line 192). In real-world production environments, big companies such as OpenAI and Google, the volume of requests is substantial and continues to increase over time. For instance, in April, ChatGPT handled approximately 1.7 billion requests, resulting in an average request rate of nearly 700 per second. This exceeds the scale of our experimental setup, where the group size was set at 256. Conversely, smaller companies with a limited number of requests can opt for stream inference instead of batch processing. Our method will become increasingly critical as AI technology continues to advance and become more widespread.
>
> We think a fairness metric for measuring individual request delay is very important. We introduce three metrics for comparison: max wait time, average wait time, and their ratio (max wait multiplier). We define the wait time for a user to be the delay from receiving the request (start of processing a group) to generate the corresponding response. With group size 256, our method saves 63% average wait time compared to the vanilla. Due to the FCR mechanism, our max wait multiplier is 2.7, which is 1.4 times the vanilla's one. However, with inference speed acceleration, the max wait time is also reduced by 48%.
>
> **Weakness-2**: Our method emphasizes that the order of requests is not a crucial factor. The key lies in how we assemble batches. Assuming that requests in a group arrive simultaneously, reordering the index has minimal additional overhead. The main inference latency is primarily influenced by the perception of response length rather than the reordering of indices.
>
> Once batches are formed, we can process them in any order. While the order of batch processing may affect the latency for individual users in our setup, the latency is constrained by the processing time of a group. Batch shuffling can be employed to mitigate any order-related impact. In a multi-GPU environment, batches containing different predicted response lengths are dispatched to different GPUs and processed simultaneously. In this scenario, there is no explicit ordering; it is solely about batch assembly.
>
> One factor influencing inference latency is the Failure Collection and Recomputation (FCR) ratio. This ratio represents the proportion of FCR samples recalculated at the end of a batch, which causes delays in processing. To assess the effect of different FCR ratios, we modify the predicted length by a constant value 'k'. A larger predicted length results in a lower FCR ratio (more tolerable). However, a long response with a short predicted length may introduce more waste generation in batch (generating 'k' more times). The experimental results, as shown in the table, demonstrate the average and maximum wait times compared to the vanilla method, with FCR time indicating the proportion of time utilized for FCR processing.
>
> | k   | Avg. | Max | FCR ratio | FCR time | Throughput (samples/s) |
> | --- | ---  | --- | --------- | -------- | ---------- |
> | -50 | 42%  | 63% | 34%       | 21%      | 1.80       |
> | -10 | **37%**  | 53% | 20%       | 20%      | 2.10       |
> | 0   | **37%**  | **52%** | 15%       | 20%      | **2.27**       |
> | +10 | 45%  | 52% | 14%       | 14%      | 2.20       |
> | +50 | 38%  | 55% | 7%        | 7%       | 2.10       |

---

> > ### Comment · Reviewer_Nixu · 2023-08-12
> >
> > Thank you for the additional experiments. The author addressed my concerns and I am raising my rating. Please include the additional results in the final manuscript.

---

> > > ### Author Response · Authors · 2023-08-14
> > >
> > > Thank you for your response. We will incorporate the additional results.

---

### Official Review · Reviewer_u5zm · 2023-07-05

**Soundness:** 3 good
**Presentation:** 2 fair
**Contribution:** 3 good
**Rating:** 6
**Confidence:** 3

**Summary:**

>**Rebuttal:** The provided details satisfy my concerns. I think this paper should be accepted after applying the agreed changes.

>**TL;DR:** The paper presents a new technique to reduce the inference time of LLMs under intensive usage. This is an important problem that can reduce wasteful computations. However, the paper is missing some key comparisons and the experimental methodology is lacking justifications. Addressing my concerns and questions would improve my score.

This paper proposes a new to reduce the inference time of LLMs under intensive usage. The technique save wasteful computations by predicting the response length and aggregating similar predicted lengths together. The paper presents an inference pipeline, which is composed of response length prediction, failure collection and recomputation (FCR), and variable batch size (VBS).

Experimental results on real-world instruction datasets using the Vicuna-7B model demonstrated an 86% improvement in throughput without sacrificing performance quality. The datasets include the Instruction-in-Wild and Alpaca. The proposed technique is compared to previous works and outperforms them on both datasets.

**Strengths:**

* **S.1.** The proposed technique can reduce inference time of LLMs and gain a 86% performance improvement.
* **S.2.** The paper tackles an important problem of reducing the LLM inference time and wasteful computations.
* **S.3.** The proposed technique outperforms previous compared works on two datasets.
* **S.4.** Reproduction code is provided as part of the submission.

**Weaknesses:**

* **W.1.** Th paper lacks comparison to existing LLM inference works such as [1][2].
* **W.2.** Some the key technique attributes are not well justified. For example, the target length prediction is four times the actual length. No experiments or ablations are provided to justify the "four". Another example is the FCR mechanism, which immediately stops the generation process when the maximum predicted length has been surpassed. There might cases where the generation process might only need a few more tokens to complete the task.
* **W.3.** The proposed technique relies on very short inputs in order to be effective. This is rarely the case for chat-bot, which require support for multi-turn conversations and are given as the examples for LLM usage. Furthermore, the proposed technique relies on high LLm usage to create batches of similar predicted lengths. This high usage is typically found in chat-bots.


[1] Yu, Gyeong-In, Joo Seong Jeong, Geon-Woo Kim, Soojeong Kim, and Byung-Gon Chun. "Orca: A distributed serving system for {Transformer-Based} generative models." In 16th USENIX Symposium on Operating Systems Design and Implementation (OSDI 22), pp. 521-538. 2022.

[2] Pope, Reiner, Sholto Douglas, Aakanksha Chowdhery, Jacob Devlin, James Bradbury, Jonathan Heek, Kefan Xiao, Shivani Agrawal, and Jeff Dean. "Efficiently scaling transformer inference." Proceedings of Machine Learning and Systems 5 (2023).

**Questions:**

* **Q.1.** The paper describes several LLM use cases such as ChatGPT, Bard, and Claude. These chat-bots require support for multi-turn conversations, which expands the conversation history. How would the proposed technique work in such cases?

**Limitations:**

The limitations of the proposed technique are described throughout the paper. The limitations include overhead of length prediction and poor compatibility with long inputs. However, the latency effects of waiting for the aggregation of batches is not discussed or evaluated.

---

> ### Author Rebuttal · Authors · 2023-08-05
>
> We sincerely thank the reviewer for the insightful comments. We are pleased to see that the reviewer acknowledges our contribution. The questions are answered below.
>
> **Weakness-1**: One significant strength of our method is its compatibility with various existing toolkits. In [2], the proposal involves selecting the best multi-dimensional partitioning techniques optimized for TPU v4 slices, which focuses on operator-level improvements and is orthogonal to our method.
>
> [1] introduces the Orca system, which selectively applies batching only to a few operations instead of processing an entire batch of requests by "batchifying" all tensor operations comprising the model. In comparison, our method is more scalable as it does not split operations and can achieve better performance with a larger group size. Moreover, our approach is simpler to implement and can be integrated into the existing inference pipeline without the need for a completely different inference system like Orca. Additionally, our method can be combined with Orca for potential further improvements.
>
> **Weakness-2**: The term "four times" might be misunderstood. It refers to the process of generating the response for each request four times and then selecting the maximum length among the four generated responses (the reason for choosing the maximum instead of the mean is explained in line 225). This process is not about predicting a target length four times the actual length. Instead, the predicted length is determined by the predictor, and the length prediction for each request is dependent on the specific sample.
>
> To relieve the concern of FCR mechanism, we modify the predicted length by a constant value 'k'. A larger predicted length (more tolerable) results in a lower FCR ratio. However, a long response with a short predicted length may introduce more waste generation in batch (generating 'k' more times). We can view the FCR ratio as a tradeoff between the above-mentioned two factors. The experimental results, as shown in the table, demonstrate the average and maximum wait times (Avg. and Max) compared to the vanilla method, with FCR time indicating the proportion of time utilized for FCR processing.
>
> | k   | Avg. | Max | FCR ratio | FCR time | Throughput (samples/s) |
> | --- | ---  | --- | --------- | -------- | ---------- |
> | -50 | 42%  | 63% | 34%       | 21%      | 1.80       |
> | -10 | **37%**  | 53% | 20%       | 20%      | 2.10       |
> | 0   | **37%**  | **52%** | 15%       | 20%      | **2.27**       |
> | +10 | 45%  | 52% | 14%       | 14%      | 2.20       |
> | +50 | 38%  | 55% | 7%        | 7%       | 2.10       |
>
> When using an accurate length predictor, it is advisable to directly utilize the predicted length with a 15% FCR ratio. This approach strikes a balance between the time spent on FCR recomputation and the time wasted during batch computation. In some cases, the generation process might require just a few more tokens to complete the task, but it is difficult to determine which response will finish or if it will finish even after generating more tokens. This concern is also the reason why the maximum length among four times generation is used, as it aims to reduce the potential FCR ratio.
>
> **Weakness-3**: We recognize that high LLM usage is essential for creating batches with similar predicted lengths in chat-bot inference. However, our method does not assume anything about the input length, and long inputs do not significantly impact our approach. In Fig. 4, we observe that the processing time for inputs (only one forward pass) is relatively small compared to the response generation process, which remains the bottleneck for chat-bot inference due to token-by-token generation and multiple forwards.
>
> Despite the Instruction-in-Wild dataset containing long length requests, our method still manages to improve throughput compared to the vanilla approach. Additionally, we acknowledge the limitation discussed in line 288 and believe that further enhancements can be made in input length scheduling. Nevertheless, we emphasize that the input processing is not the bottleneck for the chat-bot; it is primarily the response generation process that poses the greatest computational challenge.
>
> **Question-1**: In multi-turn conversations, predicting the response length for each turn is feasible based on the conversation history. The method used depends on how the conversation history is saved. If only texts are preserved with key-value (kv) keys dropped, it becomes a long input request, and it can be directly handled. On the other hand, if the system keeps the kv-cache, offloading, and compression techniques (as discussed in supplementary section C) can be employed to achieve faster responses. To further enhance the approach, implementing strategies such as limited window size and input length scheduling shows promise as future work.
>
> **Limitation**: We show the Avg. and Max waiting time compared to the vanilla method. We define the wait time for a user to be the delay from receiving the request (start of processing a group) to generate the corresponding response. With group size 256, our method saves 63% average wait time compared to the vanilla. Due to the FCR mechanism, our max wait multiplier is 2.7, which is 1.4 times the vanilla's one. However, with inference speed acceleration, the max wait time is also reduced by 48%.

---

> > ### Comment · Reviewer_u5zm · 2023-08-13
> > **Response to Rebuttal**
> >
> > Thank you for the detailed answers and results. This solves some of my concerns.
> > However, I'm not convinced regarding W.3 and partially W.1.
> >
> > * **W.1.** I'm not fully convinced that there is not a single existing algorithm that can be compared to. Adding a detailed explanation including orthogonal compatible approaches should be sufficient.
> >
> > * **W.3.** This is my main open concern. The provided Fig. 4 is computed on instruction data. The input of instruction data is usually short, and the generated outputs are usually longer that the input. This is typical for instruction based datasets, but not for real world applications. In cases where the inputs are long, the Response Length Prediction time would take a large portion of the computation time. For example, let's say the task is to give a score to a very long passages. In this example, the inputs are very long and the outputs are short (a single integer). The Response Length Prediction model would take just as long as the actual inference and thus would almost increase the total inference time by two times. This is of course an edge case, but it is an actual limitation of the paper which is not explored.
> >
> > Addressing my concerns would improve my score.

---

> > > ### Author Response · Authors · 2023-08-14
> > >
> > > Thank you for your response. Let's first discuss about your main concern **W.3**.
> > >
> > > - We discussed the reviewer's concern in the "Limitation and Discussion" section (line 288). While long input contexts have overheads, they don't predominantly appear in real-world applications. As you mentioned, the lengthy input is an edge case. For such cases, we recommend using traditional pipelines for lengthy inputs and our method for others. Specifically, since the input length is known, we can simply fallback to naive inference pipeline by setting a threshold. For instance, when the input length is longer than 512 tokens, we can always adapt naive inference instead. This would achieve a better trade-off and combine the merits of the both approachs. A future direction might merge input-length scheduling with our output length strategies for optimized inference.
> > > - Table 2 highlights that real-world chatbots like GPT-4 and Claude can perceive response length. We can thus employ the Perception in Advance (PiA) method (introduced line 95) as detailed in Appendix Section C. In this case, the length prediction and response generation model is the same one and thus the kv-cache can be reused. Using PiA's kv-cache for output creation avoids additional computational steps for very long inputs. However, without public access to these models' weights, we can't provide experiments on them.
> > > - We've covered both strategies in our paper and plan to streamline this discussion in our revisions. We argue that these limitations don't detract from our contribution, as applications with lengthy input contexts can benefit from the mentioned solutions.
> > >
> > > For your concern **W.1**, we claim that our method is orthogonal to other methods. Following your advice, we will add a detailed explanation in "Efficient LLM Inference" subsection. We discuss other inference acceleration methods by their categories according to [18]:
> > >
> > > 1. Optimization strategies such as pruning and quantization [8, 11, 35]: These reduce FLOPs needed for a forward pass. Our method does not affect this and thus is compatible.
> > > 2. Mapping and scheduling of operations [5,7]: Our approach retains the transformer's operation type and sequence, allowing existing strategies to apply.
> > > 3. Optimizing batch quantization: [10] prioritizes input length, while we focus on output length. [3] focuses in few-shot settings, a minimal usage context.
> > >
> > > As a result, most of them optimizes the inference speed in a different dimension and we think comparing our method's speedup with them bring no more insights into our paper. Combining these methods into a comprehensive system is deferred for future work, as it surpasses this paper's scope.
> > >
> > > (number reference is the same as the main text)

---

> > > > ### Comment · Reviewer_u5zm · 2023-08-14
> > > > **Response to Rebuttal #2**
> > > >
> > > > Thank you for the detailed explanations. This partially solves my concerns regarding W.1. and W.3.
> > > > * **W.1.** Although it is not ideal to not have a comparison, under the circumstances, I believe that adding the additional details and explanations should be sufficient.
> > > > * **W.3.** The Perception in Advance (PiA) is a great solution for cases with long inputs. However, PiA has a negative impact on the quality of the response. This creates an unexplored tradeoff between speed and quality. Adding more details or results on this tradeoff should eb sufficient.

---

> > > > > ### Author Response · Authors · 2023-08-15
> > > > >
> > > > > Thank you for your response. We acknowledge using PiA with API calls affects the quality of response. However, if applied correctly, the PiA can be integrated without affecting the quality of the response, requiring only minimal adjustments.
> > > > >
> > > > > To understand the PiA process, consider the following format:
> > > > >
> > > > > > **[Input] *[PiA Prompt]* *[PiA Response]* [Response]**
> > > > >
> > > > > Here, the PiA prompt might be something like "Please estimate the length of your prediction," and the subsequent PiA Response could be "The estimated response length is X". The primary overhead of this method involves the generation of the aforementioned tokens, typically no more than 30 in number. It's vital to note that both the Input and the Response can be detached from the PiA Prompt and its Response. The reason is that the Input precedes PiA parts, and the Response is solely reliant on the Input.
> > > > >
> > > > > To ensure the quality of the response remains unchanged, we can use the following two methods:
> > > > >
> > > > > 1. After the generation of the PiA response, we can exclude both the PiA Prompt and PiA Response embeddings. The generation of the primary Response would then only use the kv-cache of the input.
> > > > > 2. Alternatively, the Response can be generated with both the PiA Prompt and PiA Response masked.
> > > > >
> > > > > Both these strategies ensure the preservation of generation quality. In our experiments with models like ChatGPT and other leading-edge Large Language Models (LLMs), merely invoking the API didn't allow for the bypassing of the PiA prompt's influence. However, this implementation is quite simple for the companies. This observation also led us to introduce the Perception Only (PO) approach, wherein we compared the mean predicted length against the original prediction length as an estimation of performance of PiA method in practice.
> > > > >
> > > > > We appreciate the reviewer's insight which has added clarity to this aspect. The discussions will be added in Section 3.1 in our revision.

---

> > > > > > ### Comment · Reviewer_u5zm · 2023-08-15
> > > > > > **Response to Rebuttal #3**
> > > > > >
> > > > > > Thank you for the detailed answers.
> > > > > >
> > > > > > The provided details satisfy my concerns. I will update my review accordingly.

---

### Official Review · Reviewer_WTWP · 2023-07-06

**Soundness:** 3 good
**Presentation:** 3 good
**Contribution:** 3 good
**Rating:** 6
**Confidence:** 5

**Summary:**

This paper comes up with the technique of using LLM to help LLMs’ inference to be more efficient. It predicts the queries’ response length, and group the those with similar response length into the same micro-batch, so that the inference efficiency can be effectively improved.

**Strengths:**

In the experiments, the proposed method gains significant improvement in terms of inference speed.
It is reasonable that the token redundancy leads to inefficient inference in batch.
It is effective and easy to implement for existing LLMs.


**Weaknesses:**

The metric of horizontal axis in Fig. 2 (a) is missing.

**Questions:**

Imaging a scenario that a user’s query has extreme long response length predicted, would this user wait long time for the response?
How about to use a small model to learn to predict the response length?


**Limitations:**

None.

---

> ### Author Rebuttal · Authors · 2023-08-05
>
> We sincerely thank the reviewer for the insightful comments. We are pleased to see that the reviewer acknowledges our contribution. The questions are answered below.
>
> **Question-1**: No matter how extreme length is predicted (we can also assume a worst case: this sample actually gets a very short response), the sample is processed within a group of group size 256 in our experiment. This means the wait time for a user is bounded. We measured that our methods save 63% average waiting time for all users and 48% for the longest waiting users. Although the relative waiting time for the worst case is enlarged by 40% w.r.t. the average waiting time, our method's acceleration makes the real waiting time less for **all** users.
>
> In addition, in a group, we dispatch different samples into batches and the execution order of the batches can be shuffled, which means a sample with extreme length predicted may not be the last one for computation. The situation is even better in a multi-gpu situation. The sequence scheduling only assembles batches and the batches can be processed by different GPUs at the same time. Thus, a wrongly predicted response length will not lead to long time waiting.
>
> Using a small model can be a tradeoff between response length prediction overhead and wasted tokens saved. In Table 4, we show Pooling + MLP can improve by 61%, which is 25% lower than using the instruction-tuned model. In Table 3, a smaller model (GPT-2) achieves worse performance than the Pooling + MLP and thus may not yield better performance.
>
> **Weakness-1**: The metric of horizontal axis is "token" for Fig. 2(a) and we will add it in the revision.

---

### Official Review · Reviewer_KzBJ · 2023-07-21

**Soundness:** 3 good
**Presentation:** 3 good
**Contribution:** 2 fair
**Rating:** 6
**Confidence:** 4

**Summary:**

The authors propose to improve the throughput of the LLM inference systems by correctly predicting the length of the response.

Method summary:
1. Predict the length of the response (Binning length for prediction modules to learn better)
2. Use the prediction to batch the queries with similar prediction to improve throughput (use variable batch size to leverage GPUs while managing memory requirements)
3. Failure collection module to cut-off mispredicted batch evaluation. To ensure that this module is not triggered too often, it is advisable to over-estimate length of the prediction


**Strengths:**

1. The paper tackles an important problem, provides a simple recipe for the solution and works reasonably well.
2. The evaluation is to the point. Answers all the natural questions that might arise.

**Weaknesses:**

The writing can be better. Some details in the questions section.

**Questions:**

Comments:
1. For table two to make sense, it would be useful to have some data statistics like distribution of response lengths on those 175 instructions.
2. Table 3 caption can be improved a lot. It was not clear what i am looking at first. The table talks about vicuna model inference and various prediction methods for that. Elaborate on the caption.

Questions:
1. In table 4. why is the vanilla avg length significantly different from other lengths including ground-truth predictor?


**Limitations:**

Yes

---

> ### Author Rebuttal · Authors · 2023-08-05
>
> We sincerely thank the reviewer for the insightful comments. We are pleased to see that the reviewer acknowledges our contribution. The questions are answered below.
>
> **Question-1**: We appreciate the reviewer's clarification on the 'Avg. length' metric. Indeed, the metric does not means the average length of the generated response but rather measures the average token generation. The latter one includes wasted tokens due to short sentences waiting for the long one to be completed. In fact, since we do not change the model, with a same random seed, the response for a specific prompt under different inference acceleration methods (including vanilla, ground-truth predictor and ours) is the same and so is the response length. Therefore, a smaller Avg. length means less waste in token generation and better performance. The vanilla Avg. length is much worse than the ground-truth predictor because it has a poor performance. We will change the 'Avg. length' metric to 'Tokens wasted' metric for better understanding.
>
> **Comment-1**: The distribution of response lengths between ChatGPT and Vicuna is given in Figure 2.(a) which can be a reference.
>
> **Comment-2**: We acknowledge the need for a clearer caption for the table. In the revised version, we will modify the caption to read "Response length perception performance comparison: we evaluate different prediction method for vicuna model inference length on 10k instructions."

---

> > ### Comment · Reviewer_KzBJ · 2023-08-12
> > **Response to the authors**
> >
> > Thank you for the clarifications. Please make the discussed changes to the manuscript. Hope it gets in!

---

> > > ### Author Response · Authors · 2023-08-14
> > >
> > > Thank you for your response. We will incorporate the changes.

---

### Decision · Program_Chairs · 2023-09-21

**Decision:**

Accept (poster)

**Comment:**

The paper initially received mixed reviews. The authors' rebuttal addressed concerns that reviewers raised. The proposed idea seems to simple and effective. By predicting the response length first, the proposed method can group queries with similar output lengths to increase LLM inference throughput. All reviewers agreed that this is a good and potentially practically useful. It is expected that authors integrate the promised changes to the final version of the paper.